**Data Availability Statement:** All relevant data are within the paper and its Supporting Information files 2. Data sharing statement All patient data were

# Neonatal mortality and associated factors among neonates admitted to neonatal intensive care unit at public hospitals of Somali Regional State, Eastern Ethiopia: A multicenter retrospective analysis

**Hamda Ahmed Mohamed**[1], **Zemenu Shiferaw**[2], **Abdurahman Kedir Roble**[3], **Mohammed Abdurke Kure**[4]*

1 Department of Midwifery, Jigjiga Health Sciences College, Jigjiga, Ethiopia, 2 School of Public Health, College of Medicine and Health Sciences, Jigjiga University, Jigjiga, Ethiopia, 3 Department of Midwifery, College of Medicine and Health Sciences, Jigjiga University, Jigjiga, Ethiopia, 4 School of Nursing and Midwifery, College of Health and Medical Sciences, Haramaya University, Harar, Ethiopia

* mameelemo@gmail.com

## Abstract

### Background

Neonatal mortality remains a public health problem in the developing world. Globally, around 2.5 million neonatal deaths are reported annually with the highest mortality concentrated in sub-Saharan Africa and South Asia. In comparison with countries demonstrating the lowest neonatal mortality, the risk of mortality is over 30 times higher in sub-Saharan Africa. Ethiopia is among the countries with a high neonatal mortality rate, and the burden of this mortality remains unreported in many pastoralist areas such as Somali Regional State, Eastern Ethiopia. We aimed to investigate factors associated with neonatal mortality in public Hospitals of the Somali Regional State in Eastern Ethiopia.

### Methods

A facility-based cross-sectional study was conducted from May 1st to 30th, 2020 in three public Hospitals of Somali Regional State in Eastern Ethiopia. A total of 510 neonates admitted to neonatal intensive care units from January 2018 to December 2019 were enrolled in the study. The charts of neonates were randomly selected and retrieved. Data were collected using a pretested and validated structured questionnaire. The collected were entered into Epidata version 3.1 and exported to SPSS version 22 (IBM SPSS Statistics, 2013) for further analysis. Descriptive statistics were carried out using frequency tables, proportions, and summary measures. Predictors were assessed using a multivariable logistic regression analysis model and reported using adjusted odds ratio (AOR) with 95% Confidence Interval (CI). Statistical significance was considered at a p-value <0.05.

previously anonymized before consent was sought from the authorized bodies of the Hospitals. Data confidentiality was maintained through anonymity by removing any personal identifiers. Confidentiality of the patient information was assured by omitting their names and using card numbers instead.

**Funding:** This study was funded by Jigjiga University. The funder had no role in the study design, data collection, and analysis, decision to publish, or preparation of the manuscript.

**Competing interests:** The authors have declared that no competing interests exist.

## Results

Overall, the neonatal mortality was 18.6% [95%CI (15.31, 22.30)], equating to a rate of 186 per 1000 live births. The most common causes of mortality were prematurity (44.6%), low birth weight (33.5%), and birth asphyxia (27.6%). In the final model of multivariable analysis, predictors such as: lack of antenatal care follow-up[AOR = 3.71, 95%CI (2.13, 6.44)], neonatal sepsis [AOR = 1.84, 95%CI (1.07, 3.19), preterm birth [AOR = 2.20, 95%CI (1.02, 4.29], and birth asphyxia [AOR = 2.40, 95%CI(1.26,4.43)], and birth weight of less than 2500gms[AOR = 3.40, 95%CI(1.92, 6.01)] were statistically associated with neonatal mortality.

## Conclusion

In this study, the neonatal mortality rate was high compared to national and global targets because one in five neonates dies due to preventable causes. Modifiable and non-modifiable risk factors were identified as predictors. This result calls for all stakeholders to provide due attention to low birth weight and premature babies. Early identification and management of birth asphyxia and neonatal sepsis are also very crucial to reduce the risks of neonatal deaths.

## Introduction

Neonatal mortality (NM) continues to be a public health problem in the developing world. The first 28 days of life is the most susceptible time for children's life because they face the highest risk of dying in their first month of life [1,2]. Worldwide, about half of the global under-five mortality are due to neonatal deaths, making the neonatal period the most deadly in a child's life [3,4]. For example, in 2019, 2.4 million neonates died, with an average of 6700 deaths occurring per day in the first month of life [5,6].

Globally, most regions have made substantial progress in reducing neonatal mortality since 1990. However, the mortality rate continued to be high in developing regions. In other words, despite the global burden of neonatal deaths declined by 51%, from 5 million deaths in 1990 to 2·5 million deaths in 2017, sub-Saharan Africa(SSA) and South Asian countries have shown slow progress in reduction of the neonatal mortality. For instance, in 2017, South Asia and sub-Saharan Africa alone accounted for 79% of the global neonatal mortality [7,8]. Similarly, studies have shown that the risk of neonatal mortality is over 30 times higher in sub-Saharan Africa than in the lowest mortality country in the world [9].

Furthermore, neonatal mortality rates (NMRs) vary significantly among the countries, with a huge toll of deaths attributed to the low-income countries. For instance, in 2019, the NMR was 522/1000 in India, 270/1000 in Nigeria, 248/1000 in Pakistan, and 97/1000 in the Democratic Republic of Congo [5,8]. Ethiopia is among the countries with the highest neonatal mortality in SSA. According to the Demographic Health Survey(DHS) of 2016, in Ethiopia, the neonatal mortality rate was 29 deaths per 1000 live births, which showed a non-significant decline from 37 deaths per 1000 live births in the 2011 national survey report [10].

Furthermore, studies have shown that around 60% to 80% of neonatal deaths are attributed to prematurity and small for gestational age [5]. In addition, researchers have indicated that numerous risk factors like parents' lower educational level, lack of basic prenatal care [11], post-term pregnancy [12], low birth weight [13], congenital malformations [14], perinatal

asphyxia [15,16], preterm birth [17,18], neonatal infections [13,17,19–21], hypothermia, respiratory distress syndrome [13,22], and low Apgar score(less than 7 score) [23] are associated with neonatal deaths. Moreover, assisted vaginal delivery [20], and maternal age less than 20 years old [24] are significant predictors of neonatal deaths. Three-fourths of newborn deaths can be prevented by effective interventions such as antenatal care (ANC) during pregnancy, intrapartum care (skin-to-skin contact, early initiation of breastfeeding, newborn resuscitation, and kangaroo mother care), and postnatal care. In addition, providing the continuum of care for small and sick newborns, and good nutrition can reduce these burdens of neonatal deaths [25,26].

Ethiopia has developed various interventions since 2015 to achieve the target of sustainable development goals (SDGs). These nationally devised interventions are established to ensure the accessibility and availability of integrated management of newborns and childhood illnesses (IMNCI), neonatal intensive care units (NICU), and kangaroo mother care (KMC) at all District Hospitals. Likewise, strengthening the primary health care (PHC) services is also a national strategic plan to meet the targets of SDGs, which aimed to end preventable deaths of newborns with all countries aiming to reduce NM to at least 12 per 1,000 live-births in 2030 [27–29].

Furthermore, the Federal Ministry of Health (FMoH) has also introduced new programs such as health extension program (HEP), community health insurance program (CHIP), and community-based newborn care (CBNC) to ensure the accessibility of basic healthcare services to the rural community of Pastoralist and Semi-Pastoralist areas such as Afar and Somali regions [29,30]. With all these efforts are being implemented, still, the NMR is far from the national and global targets. Moreover, although limited studies have been conducted in Eastern Ethiopia [31–33], still, the magnitude and predictors of neonatal mortality have not been well known in the Pastoralist and Semi-pastoralist areas of the Somali Region. Therefore, we aimed to investigate the associated risk factors of neonatal mortality in selected public Hospitals of Somali Regional State, Eastern Ethiopia.

## Materials and methods

### Study setting, period, and design

A facility-based cross-sectional study (a multicenter retrospective chart review) was conducted from May 1st to 30th, 2020 in three public hospitals (Kebri-Dahar General Hospital, Dhegahbur Zonal Hospital, and Godey General hospital) in Eastern Ethiopia. Kebri-Dahar is one of the Districts in the Somali Regional State. According to the 2007 national census, the District has a total population of 136,142, of whom 77,685 are males and 58,457 females. It has one zonal hospital, one general hospital, two health centers, and four health posts. Deghbur Zonal Hospital is found in Degahbur town in the eastern part of Ethiopia, 780 Km from Addis Ababa and 171km from Jigjiga, the Regional capital. The town has an estimated total population of 42,360 of whom 9679 are reproductive age and 1339 are pregnant women. It has one Hospital, two health centers, and four health posts. Godey General Hospital is located in Godey town, 630Km from Jigjiga, Regional capital. The Godey town consists of 10 the smallest administrative unit called Kebeles. It has a total population of 75,000 (33,000 males and 42,000 females). The town has one public Hospital, one private clinic, and four health posts [34].

### Population and sampling technique

All neonates aged 0-28days admitted to NICU of selected public Hospitals of the Somali Regional State from January 2018 to December 2019 were considered as source population. All eligible and randomly selected medical records of neonates were included in the analysis.

However, medical records of neonates with incomplete and lack of pertinent information, and neonates who were initially admitted, but immediately referred to the specialized health facilities for further management were excluded from the study. The sample size of this study was calculated using the statistical software of Epi-info (Version 7.0). Thus, from the predictor variables, maternal ANC follow-up was considered as the exposure variable. This proportion was taken from a previous study report conducted in Gondar, Northern Ethiopia [20]. The proportions of neonatal deaths among mothers of a neonate who had no ANC follow-up were considered as an exposed group, and the proportions of neonatal deaths among those women who had ANC follow-up were considered as the unexposed group. Based on the above information, the following assumptions were made: Accordingly, the proportion of outcomes among unexposed group (had ANC follow-up, p = 30.8%), the proportion of outcomes among an exposed group (had no ANC follow-up, p = 13.4%), two-sided confidence level = 95%, a tolerable margin of error = 5%, power of 80%, the ratio of unexposed to exposed = 1.0, and by adding 10% contingency for non-response rate, the final sample size of the study was 514.

In the Somali Regional State, three public hospitals (Kebri-Dahar General Hospital, Degahbur Zonal Hospital, and Godey General Hospital) were purposely selected. The total number of neonates admitted to the NICU of all three Hospitals (from January 2018 to December 2019) was 830. The sample size (n = 514) was proportionally allocated to all selected public Hospitals by considering the number of admitted neonates in the last two years. Lists of medical records of neonates were taken from each Hospital and a sampling frame was developed. Medical records of neonates were selected using a simple random sampling (SRS) technique. Finally, the patients' charts were retrieved and pertinent information was obtained until the required sample size was achieved (**Fig 1**).

## Data collection tools and procedures

Data were collected using validated structured questionnaires and checklists adapted and customized from Ethiopian Demographic and Health Survey (EDHS) data collection tools [10] and developed by reviewing related literature [13,14,20]. Initially, three pediatricians and one neonatologist expert validated the content of the questionnaire. The questionnaire pretest was conducted in a similar setting and refined accordingly. Data were collected by four diploma (10+3) nurses, who had data collection experiences. Three supervisors (Bachelor of Sciences degree in nursing) were recruited and assigned to each selected Hospital for close supervision of data collectors and data collection process. Informed, voluntary, and signed consent was obtained from all authorized bodies of the selected Hospitals. All eligible medical records of neonates were manually searched from where they were previously stored and filed in the board cabinet. Eligible charts were searched, and allocated using patients' medical record numbers (MRNs). Finally, data were collected until the required sample was obtained.

## Variables and measurements

In this study, the dependent variable was neonatal mortality. This dependent variable was dichotomized as binary outcomes. Accordingly, if neonate was not survived after admission to NICU, it was recoded as "1", and if neonate was survived after admission to NICU, it was recoded as "0". Moreover, the explanatory variables were categorized as: socio-demographic variables (age of the mother, age of the neonate, sex of the neonate, residence), neonatal related factors(neonatal respiratory distress, Apgar score, congenital malformation, birth asphyxia, low birth weight, neonatal sepsis, preterm birth, hypothermia, and early and late admission), and maternal related factors (parity, place of birth, ANC follow-up, home delivery, vaginal delivery, cesarean delivery, and instrumental delivery).

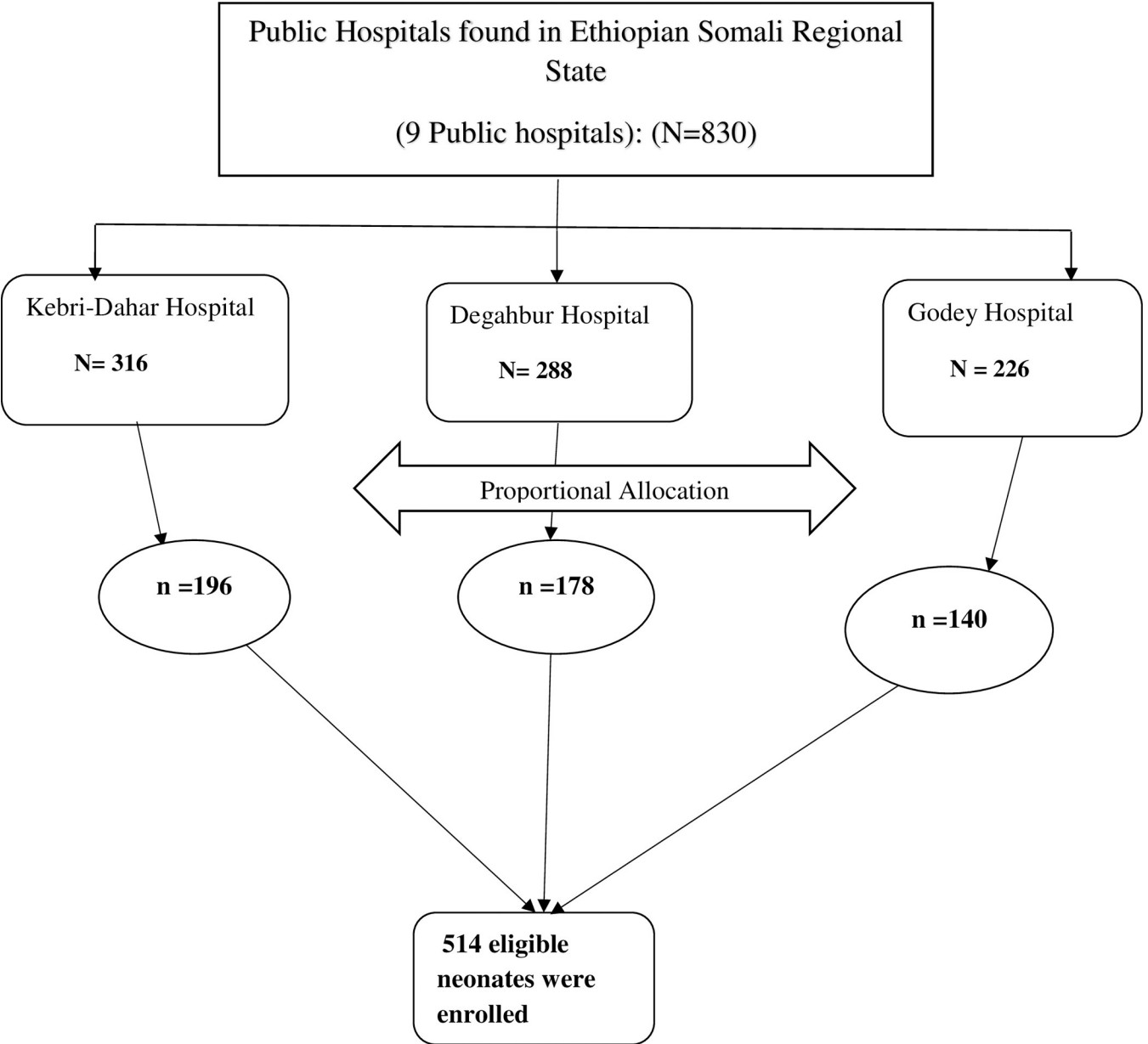

**Fig 1. Schematic representation of sampling procedures for magnitude and associated factors of neonatal mortality at selected public hospitals in Somali Regional State, Eastern Ethiopia, 2020.**

### Operational definitions

**Neonatal mortality**: the probability of dying of neonates within the first 28days of life [35]. **The magnitude of neonatal mortality**: is the proportion of neonatal death among neonates admitted to NICU. **Neonatal mortality rate**: Probability of dying during the first 28 days of life, expressed per 1,000 live births [10]. **Neonatal sepsis**: Record of infection or sepsis diagnosed either clinically or with culture by professionals during admission of the neonate and as possible causes of death and designated or recorded on the chart [12]. **Hypothermia**: defined as when the neonatal axillary temperature of less than 36.5˚c [36]. **Apgar score** is a simple evaluation system including five easily identifiable components such as heart rate, respiratory

effort, muscle tone, reflex/activity, and color. A score of 0, 1, or 2 is assigned to each component, and the sum of scores of the five components is the total score [37]. **Prematurity:** defined as when a live-born neonate is delivered before 37 completed weeks of gestation [20]. **Birth asphyxia**: is diagnosed whenever a neonate had an Apgar score $\leq 6$ at the 1[st] and 5[th] minutes and did not cry immediately after birth; had respiratory distress, and loss of neonatal reflexes [20]. **Antenatal care (ANC):** If the pregnant woman attended the ANC unit during her pregnancy at least once. **ANC follow-up**: history of one to four antenatal follow-up during current or index pregnancy at any health facility for pregnancy check-ups and chart designation or recording [38].

## Data quality control

Data collectors and supervisors were trained for two days. The main contents of the training were the following: the purpose of the study, data collection procedures, and data handling and storage. The pretest was conducted on 25 samples (5% of the total samples) of the questionnaire in Warder Hospital to ensure the validity of the tool, and the correction was made accordingly. The principal investigator and supervisors were monitored and supervised the data collection process. Data were checked for completeness and consistency, and any missed data or blank were sent back to the data collectors for correction. was conducted before data entry and analysis. The collected data were entered by two independent data clerks. Simple frequencies were run to check any missing values and outliers and crosschecked with hard copies of the questionnaire before further analysis.

## Data processing and analysis

The collected data were cleaned, coded, and entered into Epidata version 3.1 and exported to SPSS version 22 (IBM SPSS Statistics, 2013) for further analysis. Descriptive statistics were computed using frequency tables, proportion, and summary measures. Bi-variable logistic regression analysis was carried out to identify the association between each independent variable and neonatal mortality. All variables having a p-value $\leq 0.25$ in the bi-variable analysis were included in the final model of multivariable based on the assumption of selection criteria. Multi-collinearity was checked using variance inflation factor (VIF) and tolerance. Hosmer-Lemeshow goodness of fitness test was used to check for model fitness and the result was found to be insignificant (p-value $_=$ 0.501, $R^2 = 0.241$) which indicates the model was fitted. The multivariable logistic regression analysis was performed to identify the true effects of the selected predictor variables on neonatal mortality. In the final model of multivariable logistic regression analysis, the adjusted odds ratio (AOR) with 95% confidence interval (CI) were computed to estimate the true effect of independent variables on the outcome variable. The level of statistical significance was declared at a p-value <0.05.

## Ethical consideration and consent to participate

The ethical clearance was approved by the Institutional Health Research Review Committee (IHRRC) of the College of Medicine and Health Sciences, Jigjiga University (Ref.IHRRC0012/ 2020). Supportive letters were written to all three public Hospitals of the study sites. Hence, all data were previously anonymized, no informed consent was sought from the participants (mothers of neonates). Medical records of neonates who sought treatment from January 2018 to December 2019 were selected. These medical records of neonates were manually searched, and accessed from May 1[st] to 30[th], 2020. Confidentiality of the patient information was assured by omitting their names and using card numbers instead.

## Results

### Socio-demographic characteristics of the study participants

A total of five hundred fourteen (514) charts of the neonates who were admitted to the NICU of selected public Hospitals were retrieved and five hundred ten (510) were successfully extracted making the response of 99%. Four charts were excluded from the analysis because of incompleteness and lack of pertinent information. The mean age of mothers of the neonates was 26.81 years (SD = ±6.226) ranged from 16 to 42 years. Four-hundred-fifty-two (88.8%) of the mothers' neonates were from a rural setting. The mean age of neonates was 2.43days (± 4.019) and ranged from one day to twenty-seven days. More than two-thirds of the neonates (480, 94.1%) were aged 0-7days (**Table 1**).

### Maternal and neonatal related factors

The findings of this study revealed that more than half of the neonates 318(62.4%) were of normal birth weight of 2500-4000grams with a mean weight of 2909.22grams (SD = ±784.685) while around 161(31.6%) neonates were low birth weights (less than 2500gms). Concerning neonatal Apgar score, more than half of the neonates, 287(56.3%) had low Apgar scores (less than 7 scores), and more than two-thirds (64.7%) had neonatal sepsis. Around 213(41.8%) and 87(17.1%) of the neonates had a history of respiratory distress syndrome and birth asphyxia respectively. More than three-fourths of the neonates, 402(78.8%) were admitted within the first 24hoursrs (one day) and the majority, 455(89.2%) were term neonates (born between 37–42 weeks). Concerning maternal-related factors, a majority of the mothers were multipara 421 (82.5%), and more than half, 275(53.9%) of them were not utilized antenatal care during current pregnancy and only 74(29.7%) of them had more than three ANC visits. The majority, 456(89.4%) of the neonates were delivered through spontaneous vaginal delivery (SVD) followed by operative delivery 54(10.6%). Regarding the place of delivery, a vast majority of the neonates, 478(93.7%) were given birth at health institutions (**Table 2**).

### The magnitude of neonatal mortality

In this study, the magnitude of neonatal deaths in selected public Hospitals in the Somali Regional State was found to be 18.6% [95%CI (15.3, 22.3)], equating to a rate of 186 deaths per 1000 live births. In contrast, of 510 neonates admitted to NICU of selected public Hospitals of the Somali Region who were enrolled in the study, four hundred fifteen neonates (81.4%) were discharged alive (**Fig 2**).

**Table 1. Socio-demographic characteristics of neonates admitted to NICU of selected public Hospitals in Somali Regional State, Eastern Ethiopia, 2020.**

| Characteristics | Category | Frequency (n) | Percentage (%) |
|---|---|---|---|
| Age of mother | <20 years | 119 | 23.3 |
| | 20–35 years | 332 | 65.1 |
| | >35 years | 59 | 11.6 |
| Maternal residence | Rural | 452 | 88.8 |
| | Urban | 58 | 11.2 |
| Age of neonate | 0–7 days | 480 | 94.1 |
| | 8–28 days | 30 | 5.9 |
| Sex of neonate | Male | 285 | 55.9 |
| | Female | 225 | 44.1 |

**Table 2. Maternal and neonatal related factors among mothers and neonates who admitted to NICU of the selected public Hospitals in Somali Regional State, Eastern Ethiopia, 2020.**

| Characteristics | Categories | Frequency (n) | Percentage (%) |
|---|---|---|---|
| Birth weight | Low BW (<2500gm) | 161 | 31.6 |
| | Normal BW (2500-4000gm) | 318 | 62.4 |
| | Macrosomia (>4000gm) | 31 | 6.0 |
| Apgar score | <7 score | 287 | 56.3 |
| | ≥7 score | 191 | 37.5 |
| | Unknown | 32 | 6.2 |
| Neonatal sepsis | Yes | 330 | 64.7 |
| | No | 180 | 35.3 |
| Preterm birth | Yes | 56 | 11.0 |
| | No | 454 | 89.0 |
| Hypothermia | Yes | 44 | 8.60 |
| | No | 466 | 91.4 |
| Birth asphyxia | Yes | 87 | 17.1 |
| | No | 423 | 82.9 |
| Respiratory distress | Yes | 213 | 41.8 |
| | No | 297 | 58.2 |
| Admission time | Early admission (≤ 24hours) | 402 | 78.8 |
| | late admission (>24hours) | 108 | 21.2 |
| ANC follow-up | Yes | 235 | 46.1 |
| | No | 275 | 53.9 |
| Number of ANC visit | 1–3 visits | 208 | 85.2 |
| | >3visits | 36 | 14.8 |
| Parity | Primipara | 89 | 17.5 |
| | Multipara | 421 | 82.5 |
| Place of delivery | Home delivery | 32 | 6.3 |
| | Institutional delivery | 478 | 93.7 |
| Mode of delivery | Spontaneous vaginal delivery | 456 | 89.4 |
| | Operative delivery* | 54 | 10.6 |

**Key**: APGAR = A-Appearance, P-pulse, G-Grimace, Activity, Respiration; ANC = Antenatal care; BW = Birth Weight

* = Instrumental delivery, C/S delivery.

### Factors associated with neonatal mortality

In bi-variable analysis, all predictor variables having a p-value ≤ 0.25 were entered in the final model of multivariable analysis. Thus, from all tested predictor variables lack of ANC follow-up [COR = 3.49, 95%CI (2.09, 5.85)], low Apgar score[COR = 2.04, 95%CI(1.22, 3.40)], preterm birth [COR = 4.42, 95%CI (2.45, 7.94)], neonatal sepsis [COR = 1.66, 95%CI (1.01, 2.74)] and birth asphyxia [COR = 1.88, 95%CI (1.10, 3.22)], birth weight of less 2500gm(COR = 4.10, 95%CI(2.53, 6.59)] were significantly associated with neonatal mortality (**Table 3**).

In the final model of multivariable logistic regression analysis, predictor variables such as having no ANC follow-up, preterm birth, neonatal sepsis, and birth asphyxia, and low birth weight were remained statistically associated with neonatal mortality. Accordingly, the odds of neonatal mortality were 3.71 times higher among mothers who had no ANC follow-up during pregnancy compared to those neonates whose mothers had ANC follow-up during their current pregnancy[AOR = 3.71, 95%CI (2.13, 6.44)]. Preterm neonates (born before 37weeks) were 2.2 times more likely to die compared to term neonates (born after 37 weeks of pregnancy) [AOR = 2.20, (95%CI (1.02, 4.29)]. The odds of neonates who had sepsis were nearly

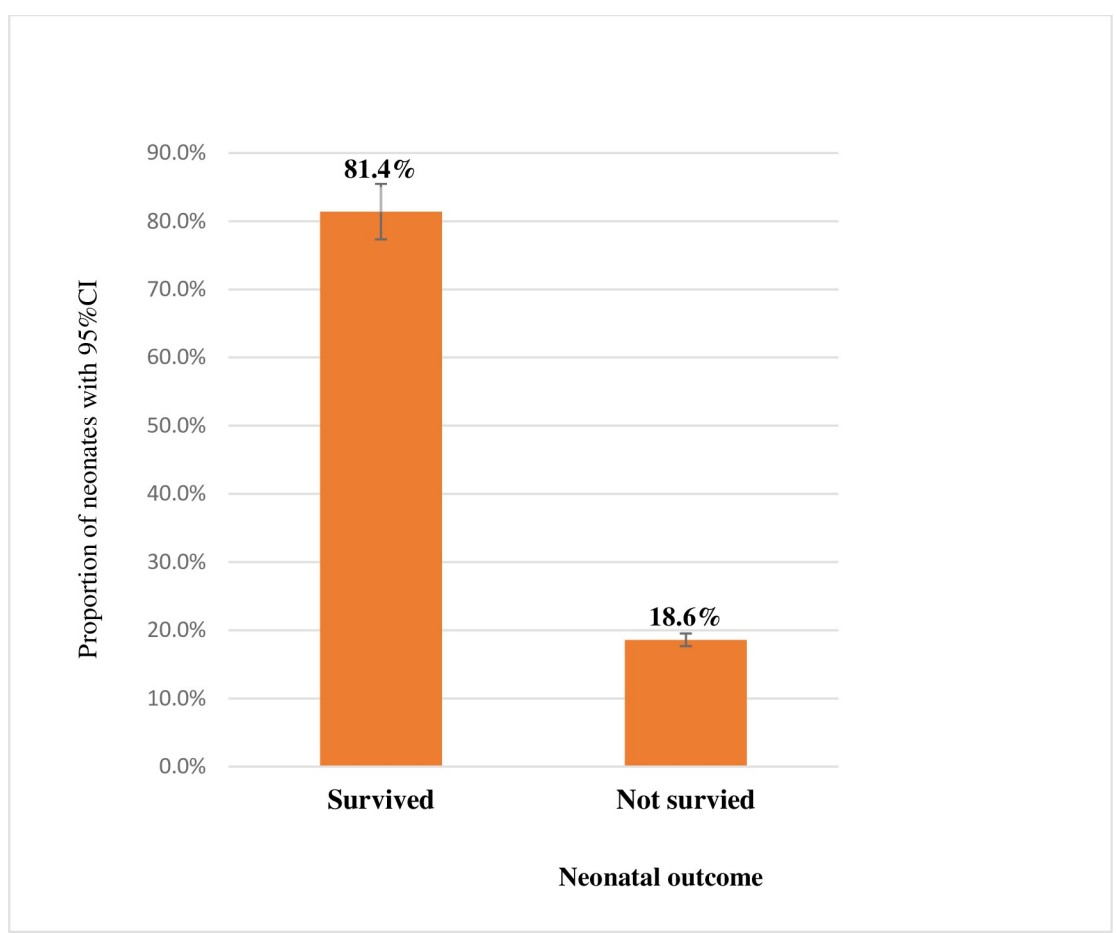

**Fig 2. Magnitude of neonatal mortality among neonates admitted to neonatal intensive care unit in selected public hospitals in Somali Regional State, Eastern Ethiopia, 2020.**

two times [AOR = 1.84, 95%CI (1.07, 3.19)] have a higher risk of death compared to those who had not sepsis during the first month of life. Concerning birth asphyxia, those neonates who had a history of birth asphyxia were 2.41'times [AOR = 2.41, 95%CI (1.26, 4.43)] more likely to die compared to those who had not asphyxia at the time of birth. Finally, the odds of neonatal mortality were more than three times higher among neonates whose birth weight was less than 2500gm compared to normal birth weight [AOR = 3.40, 95%CI(1.92, 6.01)] (Table 4).

## Discussion

In this study, the magnitude of neonatal mortality was 18.6% (equating to a rate of 186 deaths per 1000 live births). Besides, predictor variables like lack of ANC follow-up, prematurity, neonatal sepsis, birth asphyxia and low birth weight were identified as risk factors of neonatal mortality.

In this study, the proportion of neonatal is relatively comparable with previous studies conducted in Jigjiga Referral Hospital(20.5%) [31], Arba Minch General Hospital (20.2%) [39], Hawassa referral Hospital (16.5%) [15], a semi-urban Hospital in Cameroon (15.7%) [23], Democratic Republic of Congo (19.7%) [40], Ayder specialized hospital in northern Ethiopia (16.7%) [12], and Debre Markos referral Hospital in Northwest Ethiopia (21.3%) [41]. However, the current proportion of neonatal death was higher than studies conducted in different

**Table 3. Bi-variable logistic regression analysis of factors associated with neonatal mortality at selected public hospitals in Somali Regional State, Eastern Ethiopia, 2020.**

| Characteristics | Category | Neonatal outcome | | COR(CI = 95%) | P-value |
|---|---|---|---|---|---|
| | | Not survived N (%) | Survived N (%) | | |
| Age of mother (years) | ≤20 | 25(21.0) | 94(79.0) | 1.04(0.48,2.25) | 0.39 |
| | 21–35 | 58(17.5) | 274(82.5) | 0.82(0.41,1.66) | 0.580 |
| | >35 | 12(20.3) | 47(79.7) | 1 | - |
| Age of neonate (days) | 0–7 days | 89(18.5) | 391(81.5) | 0.91(0.36,2.29) | 0.840 |
| | 8–28 days | 6(20.0) | 24(80.0) | 1 | - |
| Birth weight | Low BW<2500gm | 54(33.5) | 107(66.5) | **4.10(2.53, 6.59)** | 0.0001 |
| | Normal BW (2500-4000gm) | 35(11.0) | 283(89.0) | 1 | - |
| | Macrosomia (>4000gm) | 6(19.4) | 25(80.6) | 1.94(0.74, 5.06) | 0.175 |
| Apgar score | <7 score | 65(22.6) | 222(77.4) | **2.04(1.22, 3.40)** | 0.006 |
| | Unknown | 6(18.8) | 26(81.3) | 1.61(0.60, 4.31) | 0.35 |
| | ≥7 scores | 30(13.5) | 193(86.5) | 1 | - |
| Preterm Birth | Yes | 25(44.6) | 31(55.4) | **4.42(2.45,7.94)** | 0.0001 |
| | No | 70(15.4) | 384(84.6) | 1 | |
| Neonatal sepsis | Yes | 70(21.2) | 260(78.8) | **1.67(1.01, 2.75)** | 0.044 |
| | No | 25(13.9) | 155(86.1) | 1 | - |
| Hypothermia | Yes | 8(18.2) | 36(81.8) | 0.96(0.43,2.15) | 0.94 |
| | No | 87(18.7) | 379(81.3) | 1 | - |
| Birth asphyxia | Yes | 24(27.6) | 63(72.4) | 1.68(0.97, 2.91) | 0.062 |
| | No | 71(16.8) | 352(83.2) | 1 | - |
| Residence | Rural | 75(16.6) | 377(83.4) | **0.35(1.45,4.79)** | 0.038 |
| | Urban | 21(35.1) | 37(64.9) | 1 | - |
| Respiratory distress | Yes | 47(22.1) | 166(77.9) | 1.47(0.94, 2.29) | 0.092 |
| | No | 48(16.2) | 249(83.8) | 1 | - |
| ANC follow-up | Yes | 22(9.4) | 213(90.6) | 1 | - |
| | No | 73(26.5) | 202(73.5) | **3.49(2.09,5.85)** | 0.0001 |
| Parity | Primipara | 19(21.3) | 70(78.7) | 1.23(0.70,2.16) | 0.47 |
| | Multipara | 76(18.1) | 345(81.9) | 1 | - |
| Place of birth | Home | 5(15.6) | 27(84.4) | 0.79(0.29,2.13) | 0.65 |
| | Health facility | 90(18.8) | 388(81.2) | 1 | - |
| Mode of delivery | SDV | 81(17.8) | 375(82.2) | 1 | - |
| | Operative delivery | 14(25.9) | 40(74.1) | 0.61(0.32,1.18)* | 0.15 |

**Key**: *APGAR: A-Appearance, P-pulse, G-Grimace, Activity, Respiration; SVD: Spontaneous Vaginal Delivery, ANC: Antenatal care, BW = Birth weight, LBW: Low Birth Weight, COR = Crude Odds Ratio.*

parts of the world such as Karamara General Hospital(5.7%) [22], Jimma Referral Hospital, southwest Ethiopia(13.3%) [14], Felege Hiwot referral Hospital, Northern Ethiopia (13.29%) [42], and Asmara city, Eritrea [13]. The possible justification for this discrepancy might be the difference in socio-demographic characteristics of the study participants. Moreover, health service accessibility and exposure to information might be lower in the former study population because the current study participants were from a highly marginalized pastoralist community in Eastern Ethiopia.

In contrast, the finding of this study is lower than the neonatal mortality reported in the Ashanti region of Ghana (51.8%) [43], Bench Maji Zone of South-West Ethiopia (22.8%) [44], and Hospital-based study in Egypt (58.8%) [45]. The difference is might be due to the methods

**Table 4. Multivariable logistic regression analysis of factors associated with neonatal mortality among neonates admitted to NICU of selected public hospitals in Somali Regional State, Eastern Ethiopia, 2020.**

| Characteristics | Categories | Neonatal outcome | | COR(CI = 95%) | AOR(CI = 95%) | P-values at AOR |
|---|---|---|---|---|---|---|
| | | Not survived N (%) | survived N (%) | | | |
| ANC follow up | Yes | 22(9.4) | 213(90.6) | 1 | 1 | - |
| | No | 73(26.5) | 202(73.5) | 3.49(2.09,5.85) | **3.71(2.13, 6.44)** | 0.0001 |
| Birth weight | Low BW(<2500gm) | 54(33.5) | 107(66.5) | 4.10(2.53, 6.59) | **3.40(1.92, 6.01)** | 0.0001 |
| | Normal BW (2500-4000gm) | 35(11.0) | 283(89.0) | 1 | **1** | - |
| | Macrosomia (>4000gm) | 6(19.4) | 25(80.6) | 1.94(0.74, 5.06) | 1.82(0.67, 4.94) | 0.24 |
| preterm birth | Yes | 25(44.6) | 31(55.4) | 4.42(2.45,7.94) | **2.20(1.02, 4.29)** | 0.04 |
| | No | 70(15.4) | 384(84.6) | 1 | 1 | - |
| Respiratory distress | Yes | 47(22.1) | 166(77.9) | 1.46(0.93,2.29) | 1.32(0.79, 2.15) | 0.30 |
| | No | 75(16.6) | 377(83.4) | 1 | 1 | - |
| Neonatal sepsis | Yes | 70(21.2) | 260(78.8) | 1.66(1.01,2.74) | **1.84(1.07, 3.19)** | 0.03 |
| | No | 25(13.9) | 155(86.1) | 1 | 1 | - |
| Apgar score | <7 score | 65(22.6) | 222(77.4) | 2.04(1.22, 3.40) | 1.54(0.87, 2.73) | 0.140 |
| | Unknown | 6(18.8) | 26(81.3) | 1.61(0.60, 4.31) | 1.26(0.42, 3.79) | 0.68 |
| | ≥7 scores | 30(13.5) | 193(86.5) | 1 | 1 | - |
| Residence | Rural | 75(16.6) | 377(83.4) | 0.35(1.45,4.79) | 0.26(0.44, 2.07) | 0.34 |
| | Urban | 21(35.1) | 37(64.9) | 1 | 1 | - |
| Birth asphyxia | Yes | 24(27.6) | 63(72.4) | 1.68 (0.97,2.91) | **2.41(1.26, 4.43)** | 0.007 |
| | No | 71(16.8) | 352(83.2) | 1 | 1 | - |
| Mode of delivery | SVD | 81(17.8) | 375(82.2) | 1 | 1 | - |
| | OD | 14(25.9) | 40(74.1) | 0.61 (0.32,1.18) | 1.61(0.77, 3.36) | 0.20 |

**Key**: *BW = Birth weight, SVD = Spontaneous Vaginal Delivery, OD = Operative Delivery, COR = Crude Odds Ratio, AOR = Adjusted Odds Ratio, 1 = Reference categories.*

of assessment and the difference in the sample size of the study. In addition, the difference in estimates might be attributed to the time gap between study periods, geographical setting of the study population. Another possible explanation is because currently the government is increasing the number of health extension workers in the rural community and introducing community health insurance programs that are motivating communities towards health services utilization.

In the final model of this study, we found that lack of ANC follow-up during pregnancy was significantly associated with neonatal mortality. Thus, those neonates whose mothers had no history of ANC follow-up during pregnancy were 3.71 times more likely to die compared to neonates born from mothers who had ANC follow-up during their pregnancy. This finding is consistent with several studies conducted in different parts of Ethiopia such as Debre-Marcos, Northern Ethiopia [41], and Hawassa Southern Ethiopia [15]. The possible explanation is because women who have not antenatal care follow-up during pregnancy are at risk of developing complications related to pregnancy and childbirth that can put the newborn at risk of death during the first month of life. In contrast, women with adequate antenatal care visits have a better chance of early detection and management of birth-related problems. This is also supported by the scientific finding of different kinds of literature that recommend ANC utilization is helpful in the identification of risk pregnancy, management of pregnancy-related complications, or prevention and treatment of concurrent diseases.

Likewise, early onset of neonatal sepsis was found to be an independent predictor of neonatal mortality. Thus, the odds of neonatal mortality were nearly two times higher among

neonates who had early onset of neonatal sepsis compared to those who had not asepsis. This result is incongruent with the previous hospital-based studies conducted in the capital city of Addis Ababa, central Ethiopia [46], and Cameroon [23]. It is also supported by a previous research report from Kersa Demographic Health Surveillance site in Ethiopia [32] in which a higher proportion of neonatal deaths was observed among neonates admitted with sepsis. The possible explanation could be justified by those neonates who had sepsis in the neonatal period are at risk of dying in the first of month life because their immunity can be extensively affected by disease progress.

Moreover, in this study, neonatal birth-asphyxia was statistically associated with neonatal mortality. Thus, the odds of neonatal mortality were 2.41 times higher among neonates admitted with birth asphyxia compared to those neonates who were not asphyxiated. This result is also supported by a study conducted in Jimma, Northern Ethiopia which indicated higher odds of death among neonates who had birth asphyxia compared to their counterparts (those neonates had no asphyxia) [14]. Similar findings were also reported from two Hospitals (Gondar Hospital and Ayder referral Hospital) in Northern Ethiopia, in which a higher odds of neonatal deaths was reported in asphyxiated newborns [12,20]. The possible explanation might be because the difficulty of breathing at the time of birth can lead to neonatal hypoxia as a result, neonates can be died because of oxygen deficiency.

Furthermore, prematurity was also independently associated with neonatal mortality. Accordingly, the odds of neonatal deaths were more than two times higher among preterm babies compared to term neonates. This also is in line with studies conducted in a semi-urban Hospital in Cameroon [17] and Guadalajara of central Mexico [47]. The possible explanation is because preterm babies had suppressed immune systems and other body defense mechanisms so that they can easily be exposed and infected with a bacterial infection. In addition, unlike term babies, preterm neonates are also prone to develop respiratory distress syndrome because their lungs are not matured like that of term babies. Finally, low birth weight was also independently associated with neonatal mortality. Thus, the odds of neonatal deaths were 4.1 times higher among neonates whose birth weight was less than 2500gms compared to normal birth weight. This finding is also supported by studies conducted elsewhere [12,13], in which high proportion of neonatal deaths was observed in low birth weight babies. The possible reason is that low birth weight babies are at a greater risk of neonatal sepsis than normal birth weight newborns because they are highly susceptible to bacterial infections. In addition, small size babies are at increased risk of hypoglycemia because of poor feeding, and even, they can easily develop hypothermia because of their susceptibility to cold environments. All these can increase the risk of neonatal deaths in low birth weight babies.

### Limitation of the study

Since we used a chart review cross-sectional study design, no causal association could have been made. The data were collected from a secondary source; some independent variables could have been missed. In addition, the study was conducted only in public health institutions; neonates who were admitted to private health facilities were not included in the study.

### Conclusion

In conclusion, the magnitude of neonatal mortality was unacceptably high compared to national and global targets. Lack of ANC follow-up during pregnancy, neonatal sepsis, preterm births, birth asphyxia and low birth weights were positively and statistically associated with neonatal mortality. Therefore, this result calls for all stakeholders to give due consideration to mitigating this neonatal mortality, especially in the Pastoralist community. In addition, due

attention should be given to low birth weight and premature babies. Healthcare providers and other stakeholders should also give more emphasis on early identification and management of birth asphyxia, and early onset of neonatal sepsis to reduce risks of neonatal deaths. Moreover, further studies such as longitudinal prospective studies are needed to identify the true effects of factors associated with neonatal mortality.

## Supporting information

**S1 Dataset. Dataset used for the analysis of Neonatal mortality (SPSS.data).**
(SAV)

## Acknowledgments

The authors thank the data collectors, clinical staff, and administrative staff of Kebri-Dahar General Hospital, Dhegahbur Zonal Hospital, and Godey General hospital for their unreserved support for this research paper and without them, this work would not be realized.

## Author Contributions

**Conceptualization:** Hamda Ahmed Mohamed, Abdurahman Kedir Roble, Mohammed Abdurke Kure.

**Data curation:** Hamda Ahmed Mohamed.

**Formal analysis:** Hamda Ahmed Mohamed, Zemenu Shiferaw, Abdurahman Kedir Roble, Mohammed Abdurke Kure.

**Funding acquisition:** Hamda Ahmed Mohamed, Zemenu Shiferaw, Abdurahman Kedir Roble, Mohammed Abdurke Kure.

**Investigation:** Hamda Ahmed Mohamed, Zemenu Shiferaw, Abdurahman Kedir Roble, Mohammed Abdurke Kure.

**Methodology:** Hamda Ahmed Mohamed, Zemenu Shiferaw, Abdurahman Kedir Roble, Mohammed Abdurke Kure.

**Project administration:** Hamda Ahmed Mohamed, Zemenu Shiferaw, Abdurahman Kedir Roble, Mohammed Abdurke Kure.

**Resources:** Hamda Ahmed Mohamed, Zemenu Shiferaw, Abdurahman Kedir Roble, Mohammed Abdurke Kure.

**Software:** Hamda Ahmed Mohamed, Zemenu Shiferaw, Abdurahman Kedir Roble, Mohammed Abdurke Kure.

**Supervision:** Hamda Ahmed Mohamed, Zemenu Shiferaw, Abdurahman Kedir Roble, Mohammed Abdurke Kure.

**Validation:** Hamda Ahmed Mohamed, Zemenu Shiferaw, Abdurahman Kedir Roble, Mohammed Abdurke Kure.

**Visualization:** Hamda Ahmed Mohamed, Zemenu Shiferaw, Abdurahman Kedir Roble, Mohammed Abdurke Kure.

**Writing – original draft:** Hamda Ahmed Mohamed, Mohammed Abdurke Kure.

**Writing – review & editing:** Zemenu Shiferaw, Abdurahman Kedir Roble, Mohammed Abdurke Kure.

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
