## [Decision Letter · Decision Letter 0]

27 May 2021

PONE-D-21-13372

Neonatal mortality and Associated Factors among Neonates Admitted to Neonatal Intensive Care Unit in Public Hospitals of Somali Regional State, Eastern Ethiopia: A two years’ retrospective Analysis

PLOS ONE

Dear Dr. Abdurke Kure

Thank you for submitting your manuscript to PLOS ONE. After careful consideration, we feel that it has merit but does not fully meet PLOS ONE’s publication criteria as it currently stands. Therefore, we invite you to submit a revised version of the manuscript that addresses the points raised during the review process.

ACADEMIC EDITOR:

Your manuscript was reviewed by 3 experts in the field. The majority identified many important problems in your submission and provided copious comments.  

Please revise the manuscript as per the reviewers' comments especially the  results section was difficult to follow. 

Please consider the reviewers' comments and provide point-by-point responses.

Please upload the figures again. 

We have noticed that you did not upload your data,  Plos one allows data to be available upon request if there are legal or ethical restrictions on sharing data publicly. For information on unacceptable data access restrictions, please see http://journals.plos.org/plosone/s/data-availability#loc-unacceptable-data-access-restrictions.

Please submit your revised manuscript by  July 10 2021 11:59PM. If you will need more time than this to complete your revisions, please reply to this message or contact the journal office at plosone@plos.org. Please include the following items when submitting your revised manuscript:

We look forward to receiving your revised manuscript.

Kind regards,

Ammal Mokhtar Metwally, Ph.D (MD)

Academic Editor

PLOS ONE

Journal Requirements:

2. In the ethics statement in the manuscript and in the online submission form, please provide additional information about the patient records used in your retrospective study, including: a) whether all data were fully anonymized before you accessed them; b) the date range (month and year) during which patients' medical records were accessed; c) the date range (month and year) during which patients whose medical records were selected for this study sought treatment. If the ethics committee waived the need for informed consent, or patients provided informed written consent to have data from their medical records used in research, please include this information.

[This study was funded by Jigjiga Universityand the Ethiopian Ministry of Science and Higher Education. The funding organizations had no role in the study design, data collection, data analysis, and writing up of the manuscript.]

 [The funders had no role in study design, data collection and analysis, decision to publish, or preparation of the manuscript.]

Reviewers' comments:

Reviewer's Responses to Questions

**Comments to the Author**

1. Is the manuscript technically sound, and do the data support the conclusions?

Reviewer #1: Yes

Reviewer #2: Yes

Reviewer #3: Partly

2. Has the statistical analysis been performed appropriately and rigorously? 

Reviewer #1: Yes

Reviewer #2: Yes

Reviewer #3: Yes

3. Have the authors made all data underlying the findings in their manuscript fully available?

Reviewer #1: Yes

Reviewer #2: Yes

Reviewer #3: Yes

4. Is the manuscript presented in an intelligible fashion and written in standard English?

Reviewer #1: No

Reviewer #2: Yes

Reviewer #3: Yes

5. Review Comments to the Author

Reviewer #1: Thank you for the opportunity to read your paper examining predictors and incidence of neonatal mortality in Eastern Ethiopia.

I read your paper with interest and think examining these cohorts, especially in the developing world, is important. Generally, I believe the conclusions are fair, albeit limited by the cross-sectional design.

My major comment is with respect to “Reviewer Question 4” which relates to clear, correct and unambiguous language, without typographical or grammatical errors.

Generally, given multiple errors in syntax & frequent repetition, the manuscript is difficult to read and as such the specifics around methodology are difficult to follow. There are multiple paragraphs of information that could readily be condensed into 2-3 sentences. I have included some of my feedback below, but this is not all-encompassing as these errors are present throughout the manuscript.

General comments:

1. A punchier title may be considered – perhaps revise to “Neonatal mortality and Associated Factors among Neonates Admitted to the Neonatal Intensive Care Unit in Eastern Ethiopia”.

2. In tables, precise p values should be stated instead of <0.25 or <0.05.

3. Figure 2 is likely unnecessary as it represents only a binary pie chart.

4. Generally, I’d suggest removing “died” and replacing with “mortality” where possible.

Introduction

5. Line 23: Insert “a” between “remains” and “public health challenge”

6. Line 25-26: Suggest replace sentence with “In comparison with countries demonstrating the lowest neonatal mortality, the risk of mortality is over 30 times higher in Sub-Saharan Africa”.

7. Line 35-36 can be simplified to “identify the effect of predictors on mortality”

8. Abstract results – suggest include p values

9. Line 52-53 in which year? 2017? Consider rewording

10. Line 63 – neonatal mortality in India and Pakistan not relevant given not in Africa. Listing rates in these countries is not necessary in this sentence. Also, is this intended to be 38 / 1000 live births, or 38% of neonatal admissions? The sentence reads as all-cause neonatal mortality.

11. Line 72 change declined to decline and clarify “but not met national target” as this doesn’t read well & doesn’t make sense here.

12. Line 83-90 from editorial point of view try to delete some of the “for instance” and “for example”

13. The introduction is overall a bit wordy and repetitious and could be shortened.

Methods

14. Need to define “Woredas” and “Kebeles” for international readers.

15. Line 109 – delete “and”

16. Line 114 – “woreda” is not capitalised as previously done.

17. Line 122 delete “and”

The methods are particularly difficult to follow given multiple typographical and grammatical errors. I have listed only a few of them above and would suggest a thorough proof reading and rephrasing in many places for clarity.

Results

18. Line 242 – replace “in” with of”

19. Again multiple typographical errors (including above) which should be proof read and corrected.

Discussion

The discussion revolves primarily around comparisons with mortality in other regions and countries. Line 289 – 315 is entirely dedicated to comparator cohorts, of which > 10 are listed. Consider abbreviating this list and condensing this to one paragraph discussing similar cohorts.

Minor comments:

20. Line 294 – How is Turkey relevant here? Suggest removing.

21. Line 301-302 remove the assertion regarding difference due to time of the study and sample size as this is stated in the following sentence anyway.

22. Line 351 delete “Strengths of the study” and “Limitation of the study”. This section needs revising. Standardised checklists and questionnaires are not necessarily a strength.

23. Lines 352-354 is poorly worded.

Conclusion

Conclusions are appropriate but again need to be reworded for simplicity.

While I think the results of this study are interesting and relevant to both the scientific community in Ethiopia and the international community, this paper requires major revision, primarily from a language point of view to reduce repetition, simplify the methodology for the reader, and reduce the burden of language errors.

Reviewer #2: The authors studied predictors of neonatal mortality in 3 public hospitals in the Somali region, Eastern Ethiopia. The research is original, it was the first in this region which is mainly a rural area. The data presented supported the conclusion and fitted with the study objective. The sample size was calculated appropriately, the sample was randomly selected and the satatistical analysis were rigorously performed. The results were properly discussed and the limitations of the study were mentioned.

However, I have some suggestions to improve the manuscripts :

1. There are some grammer and spelling mistakes

2. The p-value of the Hosmer-Lemeshow goodness of fitness test is a result and should be included with the table of the final model in the results section with the determination coefficient R2.

3. I suggest to present the final model only of the multivariable regression with the goodness of fit criteria.

Reviewer #3: The article provides a very nice picture on neonatal mortality and relevant risk factors for neonatal mortality in Ethiopia. It is worthwhile to be published. However, the article is not written carefully, abbreviations are not introduced, and country specific situation are not explained for an international readership and one citation is misleading.

General remarks:

Abbreviation need to be introduced once when used first time, and then this abbreviation shall be used through the whole text. I believe NMR is introduce three times, however ANC, NM, and NICU is not explained at all. Due to the tremendous use of abbreviation, I also suggest providing an abbreviation list where all abbreviations are explained. Please check the correct use of all abbreviation in the text.

Introduction:

The introduction explains the importance of the research question and uses relevant literature. However, the rates from neonatal mortality seems to be wrong, (line 61 ff: information of neonatal mortality is normally given in number per 1000 and not in %; the relevant cited source is connected to under 5 years mortality which does not make sense in this context; and the numbers are not repeatable. Please correct these numbers and cite a correct source.

Furthermore, I suggest introducing shortly the Apgar-score as well as ANC and ANC follow up in the introduction. You cannot expect that the international readership knows this words/abbreviations.

Method:

In the study population sample size, data collection and data analysis are presented carefully. However, the sample size calculation can be presented more condensed. And the coding of neonatal mortality can be dropped from the explanation.

Furthermore, under analysis it was mentioned that collinearity analysis was done, but the results were not presented. I expect a high collinearity between apgar-score and asphyxia (which is defined via apgar score) and ANC follow up and No of ANC-visits. Therefore, the results are important to present.

Result:

Result presentation is nice and carefully done.

As small remark to some tables:

• Table 2:, I suggest to present for binary variables only one line. If you know that 8.6 (n=44) have Hypothermia, then it is implicitly given that the rest do not have hyperthermia this line can be dropped from the table. However, that is relevant only for binary/dichotomous variables. On the other hand, there is one variable (number of ANC visits) which is not dichotomous/binary, but only presented in this form, clarification is necessary.

• Table 4: significant results are only marked under AOR but not under COR. Please add this. Furthermore, both abbreviations need to be introduced under the table.

• Figure 1. The arrow text line “Simple random sampling (SRS)” can be dropped. The line above contains 196+178+140 which gives exactly 514. No random sampling took place at this stage of sampling procedure.

• Figure 2 can be deleted, The results of this figure can be added in table 2

Discussion:

The discussion is nicely done, good work.

6. PLOS authors have the option to publish the peer review history of their article (what does this mean?). If published, this will include your full peer review and any attached files.

Reviewer #1: No

Reviewer #2: **Yes: **Hedia Bellali

Reviewer #3: No

---

## [Author Response · Author response to Decision Letter 0]

20 Jul 2021

Authors’ Response to the editor’s and Reviewers’ comments and Suggestions

Manuscript ID: PONE-D-21-13372

Journal: PLOS ONE

Dear Editors and Reviewers,

Thank you so much for giving us an opportunity to submit a revised draft of our manuscript entitled “Neonatal mortality and Associated Factors among Neonates Admitted to Neonatal Intensive Care Unit in Public Hospitals of Somali Regional State, Eastern Ethiopia: A two years’ retrospective Analysis” to this high visibility impact factor and peer reviewed Journal. We appreciate the time and effort that you and the reviewers dedicated to providing feedback on our manuscript. We are very grateful for the insightful comments and valuable improvements to our premature paper. We have incorporated most of the suggestions and comments made by handling editor, and reviewers. All comments and suggestions forwarded by Editor and reviewers were clearly stated and well addressed (a point-by-point to the Editor's and reviewers' comments and concerns) in the separated letter of "Response to Reviewers". All newly changes were highlighted in Red font color within the clean revised manuscript. Thank for your countless effort.

Authors’ Response to Editor’s Comments and Suggestions:

Title: Neonatal mortality and Associated Factors among Neonates Admitted to Neonatal Intensive Care Unit in Public Hospitals of Somali Regional State, Eastern Ethiopia: A two years’ retrospective Analysis

Authors: 1.Hamda Ahmed Mohamed(First author), 

 2.Zemenu Shiferaw (Co-author), 

 3.Abdurahman Kedir Roble(Co-author) 

 4.Mohammed Abdurke Kure(Co-author & Corresponding author)

To: Handling Editor(s)

From: Mohammed Abdurke Kure (Corresponding Author)

Subject: Submission of Incorporated Comments and Suggestions

First, we thank you for your constructive comments and helpful suggestions that helped us to improve and enrich our manuscript. Here under in the table below, we have pointed out how authors incorporated your valuable comments, suggestions and concerns one by one. 

1.Thank you for submitting your manuscript to PLOS ONE. After careful consideration, we feel that it has merit but does not fully meet PLOS ONE’s publication criteria as it currently stands. Therefore, we invite you to submit a revised version of the manuscript that addresses the points raised during the review process.

AUTHORS' RESPONSE:

Great! We are very happy and overjoyed. Thank you very much for giving us an opportunity to submit our revised manuscript to such legitimate and high visibility impact factor Journal (PLOS ONE). 

2.Your manuscript was reviewed by 3 experts in the field. The majority identified many important problems in your submission and provided copious comments. Please revise the manuscript as per the reviewers' comments especially the results section was difficult to follow. 

AUTHORS' RESPONSE:

Ok, accepted with thanks. We appreciate the time and effort that you and the reviewers dedicated to providing feedback on our manuscript. 

3.Please consider the reviewers' comments and provide point-by-point responses.

AUTHORS' RESPONSE:

Thanks a lot. Dear editor, the authors critically reviewed all comments and suggestions raised during review process and corrected all necessary modifications. The newly modified change were highlighted with red font color in the clean revised main manuscript.

4.We have noticed that you did not upload your data, Plos one allows data to be available upon request if there are legal or ethical restrictions on sharing data publicly. 

AUTHORS' RESPONSE:

Ok, great! No legal restrictions for our data set. We decided to upload all data sets used for analysis in this study without any restrictions. Thanks a lot!

Academic Editor’s Specific Comments (Journal Requirements)

5.Please ensure that your manuscript meets PLOS ONE's style requirements, including those for file naming. The PLOS ONE style templates can be found athttps://journals.plos.org/plosone/s/file?id=wjVg/PLOSOne_formatting_sample_main_body.pdf and

AUTHORS' RESPONSE:

Thank you so much. You are perfect. These are very important comments. Even it is the authors mandatory to stick to Journal’s format guidelines. Now, the authors addressed this critical issues based on your valuable suggestion. We downloaded all formats templates (PLOS Affiliations Formatting and Manuscript body formatting guidelines) from Journal’s Website, and critically read and corrected all necessary formatting. Newly changed and corrected were highlighted with red font color in the clean revised manuscript.

6.In the ethics statement in the manuscript and in the online submission form, please provide additional information about the patient records used in your retrospective study.

AUTHORS' RESPONSE:

Thanks. You are perfect. This valid comment was addressed based on your valuable suggestion. Newly changed and corrected was highlighted with red font color in the clean revised manuscript (on page11, Lines=226-232)

7.Please remove any funding-related text from the manuscript and let us know how you would like to update your Funding Statement.

AUTHORS' RESPONSE:

Ok, thanks. The editor is correct. We are very sorry; this is against PLOS ONE authors’ guideline. The authors critically reviewed this valid comment and corrected the necessary modification. The newly modified change was highlighted with red font color in the clean revised manuscript

8.If there are no restrictions, please upload the minimal anonymized data set necessary to replicate your study findings as either Supporting Information files. We will update your data availability statement on your behalf to reflect the information you provide.

AUTHORS' RESPONSE:

Thank you very much. In fact, this is a valid concern. We critically considered this point, and we decided to upload data set used for analysis, and we aploaded it as supplementary File(S1)

9.Your ethics statement should only appear in the Methods section of your manuscript. If your ethics statement is written in any section besides the Methods, please delete it from any other section.

AUTHORS' RESPONSE:

Thank you a lot for such a technical input. Now, authors considered the raised issue. After thoroughly and critically revised this important comment, we removed from previously it appeared in Acknowledgment part. Thanks!

End of authors’ responses for Handling Editor(s)

Authors’ Response to Reviewer 1’s Comments and Suggestions:

Title: Neonatal mortality and Associated Factors among Neonates Admitted to Neonatal Intensive Care Unit in Public Hospitals of Somali Regional State, Eastern Ethiopia: A two years’ retrospective Analysis

Authors: 1.Hamda Ahmed Mohamed(First author), 

 2.Zemenu Shiferaw (Co-author), 

 3.Abdurahman Kedir Roble(Co-author) 

 4.Mohammed Abdurke Kure(Co-author & Corresponding author)

To: Reviewer 1

From: Mohammed Abdurke Kure (Corresponding Author)

Subject: Submission of Incorporated Comments and Suggestions

First and foremost, we would like to thank you for your constructive and valuable comments and helpful suggestions that helped us to improve and enrich our premature manuscript. Here under in the table we have pointed out how authors incorporated your valuable comments, suggestions and concerns one by one. 

5. Specific Review’s Comments to the Authors

1.Reviewer #1: Thank you for the opportunity to read your paper examining predictors and incidence of neonatal mortality in Eastern Ethiopia.

Authors' Response: Thank you very much for countless effort to review our premature manuscript.

2.I read your paper with interest and think examining these cohorts, especially in the developing world, is important. Generally, I believe the conclusions are fair, albeit limited by the cross-sectional design

Authors' Response: 

Great! Thanks a lot. You are perfect. In the cross-sectional study design, the researchers cannot draw a conclusion of cause-effect relationship, thanks.

3.My major comment is with respect to “Reviewer Question 4” which relates to clear, correct and unambiguous language, without typographical or grammatical errors.

Authors' Response: 

Thank you so much. Authors critically considered this input. Authors acknowledged your efforts. We critically considered and incorporated all raised issues, comments, suggestions and concerns in this manuscript. Moreover, we thoroughly revised and edited the whole parts of our manuscript and extensively corrected all copy-editing errors in the clean revised manuscript. Authors also sent the manuscript to language expert/editor who critically reviewed, edited and corrected all language related errors made in a submitted manuscript. 

4.Generally, given multiple errors in syntax & frequent repetition, the manuscript is difficult to read and as such the specifics around methodology are difficult to follow.

Authors' Response: 

Thank very much. The reviewer is perfect. At initial submission, the manuscript was not written in concise manner. We critically considered this point. We took a long time, thoroughly and extensively revised and edited the whole manuscript both technically and editorially. The document was also arranged and organized as per journal template. Even we removed all previously written non-sense paragraphs and sentences and replace them with new texts both in ‘Introduction’ part and ‘Methods’ part. The newly modified change were highlighted with red font color in the clean revised manuscript.

5.There are multiple paragraphs of information that could readily be condensed into 2-3 sentences. I have included some of my feedback below, but this is not all-encompassing as these errors are present throughout the manuscript.

Authors' Response:

Thank you a lot. We appreciated your observation. We considered your deep concern and explicitly incorporated your input both technically and editorially. The document was newly arranged and organized, even we removed all previously written non-sense paragraphs and sentences, clauses and poorly worded text and replaced them with new texts both in ‘Introduction’ part and ‘Methods’ part. The newly modified change were highlighted with red font color in the clean revised manuscript.

6. A punchier title may be considered – perhaps revise to “Neonatal mortality and Associated Factors among Neonates Admitted to the Neonatal Intensive Care Unit in Eastern Ethiopia”.

Authors' Response:

Dear, reviewer, thank you very much. The authors could not understand this suggestion. Do you mean to modify the title of the paper? 

7.In tables, precise p values should be stated instead of <0.25 or <0.05

Authors Response:

Ok! Thank you reviewer. We really appreciate this valid observation. Is it better to put both p-values and ORs simultaneously? But currently, many researchers recommend only ORs instead of using both estimators simultaneously. Anyways, we incorporate this important suggestion based on your recommendation in the tables of bi-variable and multivariable analysis. 

8.Figure 2 is likely unnecessary as it represents only a binary pie chart.

Authors' Response:

Thank you very much. In fact, this is a valid concern. We critically considered this important suggestion, and removed previously appeared binary chart, and web substitute it with bar-chart because it is the most appropriate graphical representation for this binary outcome. 

9.Generally, I’d suggest removing “died” and replacing with “mortality” where possible.

Authors' Response:

Exactly! Thanks. It is not good to use the word “died” to indicate mortality. Based on this important suggestion, we made necessary changes throughout the whole document where possible!

INTRODUCTION

10.Line 23: Insert “a” between “remains” and “public health challenge”

Authors Response:

Thank you so much. This was grammatical error. We corrected this on page 2, line 41 of the main manuscript based on your recommendation.

11.Line 25-26: Suggest replace sentence with “In comparison with countries demonstrating the lowest neonatal mortality, the risk of mortality is over 30 times higher in Sub-Saharan Africa”.

Authors Response:

Thank you very much. The reviewer is correct. We considered this valid concern. Authors carefully considered your valid comments and modified based on your insightful suggestion on page 2, line 24-25 

12.Line 35-36 can be simplified to “identify the effect of predictors on mortality”

Authors Response:

Thank you a lot. We appreciated your observation. Now authors considered your valid concern and explicitly incorporated your input in the clean revised manuscript and highlighted with red font color.

13.Abstract results – suggest include p values

Authors Response:

Thank a lot. Dear editor, we appreciate your valid suggestion. Is it better to put both p-values and ORs simultaneously in the abstract? Most of Biomedical researchers recommend AOR with CIs,(AOR, 95%CI). In previous comment, we incorporate p-value in the tables of bi-variable and multivariable analysis. However, here if add p-value with AOR and 95%CIs, the abstract may become bulk and distorted. Abstract should be short and concise.

14.Line 52-53 in which year? 2017? Consider rewording

Authors Response:

Definitely! Thank you so much. This important comment was modified and highlighted with red font color in the clean revised main manuscript. Thanks.

15.Line 63 – neonatal mortality in India and Pakistan not relevant given not in Africa. Listing rates in these countries is not necessary in this sentence. Also, is this intended to be 38 / 1000 live births, or 38% of neonatal admissions? The sentence reads as all-cause neonatal mortality.

Authors' Responses:

OK! Thank you. Here we found 2 important comments. First. Neonata Mortality in Indian and Pakistan: Exactly, you are perfect. We are very sorry! This was unintentionally introduced error during first draft. Actually, the paragraph started with indicating both Africa and Asian regions. However, “Asia” was missed unintentionally. This was editorial error. Now, we corrected this comment on page 3, line 59-60. Thank you!

Second: NMR should be per 1000 live births. The reviewer is correct. What you are suggesting is the standards. We appreciate this valid observation. Normally, Neonatal mortality can expressed 1000 per live births Rate (NMR=per 1000 live births). However, in small-scale research, we can estimate neonatal death in “magnitude” or “proportion” to show the prevalence. Based on your recommendation, we corrected this valid comment, and corrected on page 3, line 61-66. 

16.Line 72 change declined to decline and clarify “but not met national target” as this does not read well & doesn’t make sense here.

Authors Response:

Thank you a lot. We appreciated your observation. Now authors considered your valid concern and explicitly incorporated your input in the clean revised manuscript and highlighted with red font color.

17.Line 83-90 from editorial point of view try to delete some of the “for instance” and “for example”

Authors' Response:

Thank you a lot. We appreciated your observation. Now authors considered your valid concern and explicitly incorporated your input in the clean revised manuscript and highlighted with red font color.

18.The introduction is overall a bit wordy and repetitious and could be shortened

Authors Response:

Great! Thank very much. We have corrected and rephrased based on your recommendation. Even the authors explicitly and critically reviewed this valid comment and rewrote. The previous bulk paragraphs were removed and organized in concise manner. We highlighted the change in red font in the main clean revised manuscript.

METHODS

19.Need to define “Woredas” and “Kebeles” for international readers.

Authors' Response:

Thanks. Thank you a lot for such technical input. This lacks standards, and difficult to understand by scientific community. We made the correction accordingly. (Woreda=District, Kebeles=The smallest administrative unit in Ethiopia. Thank you very much!

20.Line=109–delete“and”

Authors Response:

Great, thanks a lot. This was editorial error, and we corrected this on page 5, line 106-108 in the clean revised manuscript.

21.Line 114 – “woreda” is not capitalised as previously done.

Authors Responses:

Thank you. The authors have corrected such a language and editing errors. We corrected this grammatical based your insightful comment error here and elsewhere in the clean revised manuscript.

22.. Line 122 delete “and”

Authors Response:

Thank you so much. This was editorial and grammatical error. We corrected this on page 5, lines 124-126.

23.The methods are particularly difficult to follow given multiple typographical and grammatical errors. I have listed only a few of them above and would suggest a thorough proof reading and rephrasing in many places for clarity.

Authors Response:

Thank you a lot for such a technical input. You are perfect. Previously, there were bulky paragraphs and sentences in the methods part of this manuscript. Now, the authors considered the raised issue. After thoroughly and critically revised this important comment, we removed all previous bulky texts and long paragraphs/sentences and replaced it with newly corrected one, and all changes were highlighted with red font color in the clean revised manuscript from page 5-9. 

RESULTS

24.Line 242 – replace “in” with of”

Authors Response:

Important, Thank you. We have corrected these unnecessary editing errors based on your valuable recommendations and suggestions.

25.Again multiple typographical errors (including above) which should be proof read and corrected.

Authors Responses:

Great! Thanks a lot for such implicit and critical review for our premature paper. We are very sorry for such unnecessary full of editorial errors for the whole manuscript. Authors critically considered this input. Now, we thoroughly revised and edited the whole parts of our manuscript and extensively corrected all copy-editing errors in the clean revised manuscript. Authors also sent the manuscript to language expert/editor who critically reviewed, edited and corrected all language related errors made in a submitted manuscript.

Important, Thank you. We have corrected these unnecessary editing errors based on your valuable recommendations and suggestions.

DISCUSSION

26.The discussion revolves primarily around comparisons with mortality in other regions and countries. Line 289 – 315 is entirely dedicated to comparator cohorts, of which > 10 are listed. Consider abbreviating this list and condensing this to one paragraph discussing similar cohorts.

Authors' Response:

Thank you so much. We appreciated your observation. The authors critically reviewed this important comment. We condensed and shortened previously bulky long paragraphs based on your valuable suggestions. 

27.Line 294 – How is Turkey relevant here? Suggest removing.

AUTHORS' RESPONSE:

Thank you very much. In fact, this is a valid concern. We critically considered this point, and modified accordingly. 

28.Line 301-302 remove the assertion regarding difference due to time of the study and sample size as this is stated in the following sentence anyway.

Authors' Response:

Thanks a lot. You are perfect. This valid comment was incorporated based on your valuable suggestion. Newly changed and corrected was highlighted with Red font color in the Clean revised manuscript.

29.Line 351 delete “Strengths of the study” and “Limitation of the study”. This section needs revising. Standardized+ checklists and questionnaires are not necessarily a strength.

Authors' Response:

Thank you again. You are correct. This is editorial errors and Technical errors. We have corrected this valid concern based on your important implicit suggestion. We delete unnecessary text and highlighted with red font color in the clean revised manuscript.

30.Lines 352-354 is poorly worded.

Authors Response:

Great, Thank you so much. Very appreciable comment and suggestion. We have corrected these unnecessary poorly worded following your insightful recommendations and suggestions.

CONCLUSION

31.Conclusions are appropriate but again need to be reworded for simplicity.While I think the results of this study are interesting and relevant to both the scientific community in Ethiopia and the international community, this paper requires major revision, primarily from a language point of view to reduce repetition, simplify the methodology for the reader, and reduce the burden of language errors.

Authors Response:

Thank you very much for such valuable intellectual input. Authors critically considered this input. Now, we thoroughly revised and edited the whole parts of our manuscript and extensively corrected all copy-editing errors in the clean revised manuscript. Authors also sent the manuscript to language expert/editor who critically reviewed, edited and corrected all language related errors made in a submitted manuscript.

NB: New notification for reviewer 1

Dear, reviewer, thank you very much for your countless effort. Finally, we kindly notify you that during revision of our manuscript, we revised the final model of multivariable analysis and we checked one predictor variable (Birth weight of newborn). The BW was categorized as:

1.LBW= less than 2500mg , 2.Normal BW= 2500-4000mg

3.Macrosomia: >4000mg. In SPSS analysis, initially we used the “Macrosomia” as Ref. category, and no significant association was found in multivariable analysis. However, in current revision, the authors critically revise this issue and shifted/substituted the previous Ref. category (>4000mg) to “Normal BW(2500-400mg)”, considering the remaining two categories as exposure. As a result, LBW becomes significant predictor of Neonatal mortality. For further details, we submitted SPSS data used for analysis to the Journal as Supplementary File 1(S1). We highlighted newly modified changes with red font color in the Table 3 and 4, and elsewhere in the text. Thank you very much for your time and consideration.

End of authors responses for Reviewer 1

Authors’ Response to Reviewer 2’s Comments and Suggestions:

Title: Neonatal mortality and Associated Factors among Neonates Admitted to Neonatal Intensive Care Unit in Public Hospitals of Somali Regional State, Eastern Ethiopia: A two years’ retrospective Analysis

Authors: 1.Hamda Ahmed Mohamed(First author), 

 2.Zemenu Shiferaw (Co-author), 

 3.Abdurahman Kedir Roble(Co-author) 

 4.Mohammed Abdurke Kure(Co-author & Corresponding author)

To: Reviewer 2

From: Mohammed Abdurke Kure (Corresponding Author)

Subject: Submission of Incorporated Comments and Suggestions

First and foremost, we would like to acknowledge you for your constructive and valuable comments and helpful suggestions that helped us to improve and enrich our manuscript. Here under in the table we have pointed out how authors incorporated your valuable comments, suggestions and concerns one by one. 

1.Reviewer #2: The authors studied predictors of neonatal mortality in 3 public hospitals in the Somali region, Eastern Ethiopia. The research is original; it was the first in this region, which is mainly a rural area. The data presented supported the conclusion and fitted with the study objective. The sample size was calculated appropriately, the sample was randomly selected and the satatistical analysis were rigorously performed. The results were properly discussed and the limitations of the study were mentioned.

Authors Response:

Thank you very much. We would like to thank you for your appreciation and constructive suggestion. Further, we revised and enriched the paper after previous initial submission to the journal.

2.Reviewer #2 However, I have some suggestions to improve the manuscripts :

Authors Response:

Thank you a lot. We accepted all your valid suggestion and concern in this paper, and correct accordingly.

3.There are some grammar and spelling mistakes

Authors' Response:

Thank you so much. Authors critically considered this input. Now, we thoroughly revised and edited the whole parts of our manuscript and extensively corrected all copy-editing errors in the clean revised manuscript. Authors also sent the manuscript to language expert/editor who critically reviewed, edited and corrected all language related errors made in a submitted manuscript.

4.The p-value of the Hosmer-Lemeshow goodness of fitness test is a result and should be included with the table of the final model in the results section with the determination coefficient R2.

Authors Response:

Ok, thanks! We really appreciate this valid suggestion. Dear reviewer, is it better to put H-L goodness of fitness test result and R2 in Final model of Multivariable analysis? Where do we put this result in table? Do we create its own column? Most of biomedical researchers put the results of H-L goodness test, VIF and tolerance in methods part. Anyways, we incorporate this important suggestion in the clean revised manuscript in the method part. 

5. I suggest to present the final model only of the multivariable regression with the goodness of fit criteria.

Authors' Response:

Ok! Thank you so much, reviewer. We also put this for only multivariable logistic regression analysis.

NB: New Notification for Reviewer 2

Dear, reviewer, thank you very much for your countless effort. Finally, we kindly notify you that during revision of our manuscript, we revised the final model of multivariable analysis and we checked one predictor variable (Birth weight of newborn). The BW was categorized as:

1.LBW= less than 2500mg , 2.Normal BW= 2500-4000mg

3.Macrosomia: >4000mg. In SPSS analysis, initially we used the “Macrosomia” as Ref. category, and no significant association was found in multivariable analysis. However, in current revision, the authors critically revise this issue and shifted/substituted the previous Ref. category (>4000mg) to “Normal BW(2500-400mg)”, considering the remaining two categories as exposure. As a result, LBW becomes significant predictor of Neonatal mortality. For further details, we submitted SPSS data used for analysis to the Journal as Supplementary File 1(S1). We highlighted newly modified changes with red font color in the Table 3 and 4, and elsewhere in the text. Thank you very much for your time and consideration

End of authors responses to Reviewer 2!!!

Authors’ Response to Reviewer 3’s Comments and Suggestions:

Title: Neonatal mortality and Associated Factors among Neonates Admitted to Neonatal Intensive Care Unit in Public Hospitals of Somali Regional State, Eastern Ethiopia: A two years’ retrospective Analysis

Authors: 1. Hamda Ahmed Mohamed(First author), 

 2. Zemenu Shiferaw (Co-author), 

 3. Abdurahman Kedir Roble(Co-author) 

 4. Mohammed Abdurke Kure(Co-author & Corresponding author)

To: Reviewer 3

From: Mohammed Abdurke Kure (Corresponding Author)

Subject: Submission of Incorporated Comments and Suggestions

First and foremost, we would like to thank you for your insightful comments and helpful suggestions that helped us to improve and enrich our manuscript. Here under in the table we have pointed out how authors incorporated your valuable comments, suggestions and concerns one by one. 

1.Reviewer #3: The article provides a very nice picture on neonatal mortality and relevant risk factors for neonatal mortality in Ethiopia. It is worthwhile to be published. However, the article is not written carefully, abbreviations are not introduced, and country specific situation are not explained for an international readership and one citation is misleading.

Authors' Response:

Great, Thank you so much. Very appreciable comment and suggestion. Authors acknowledge for your constructive suggestions!

2.Abbreviation need to be introduced once when used first time, and then this abbreviation shall be used through the whole text. I believe NMR is introduce three times, however ANC, NM, and NICU is not explained at all. Due to the tremendous use of abbreviation, I also suggest providing an abbreviation list where all abbreviations are explained. Please check the correct use of all abbreviation in the text.

Authors' Response:

Thanks a lot. You are perfect. These are very important observation. These valid comments and suggestions were addressed based on your insightful suggestion in the introduction part and elsewhere. All abbreviations/Acronyms (appear at least 3 or more in the document) were critically revised and corrected based on your valuable suggestions. Newly changed and corrected were highlighted with red font color in the clean revised manuscript. 

INTRODUCTION

3.-The introduction explains the importance of the research question and uses relevant literature. However, the rates from neonatal mortality seems to be wrong, (line 61 ff: information of neonatal mortality is normally given in number per 1000 and not in %; the relevant cited source is connected to under 5 years mortality which does not make sense in this context; and the numbers are not repeatable. Please correct these numbers and cite a correct source.

Authors' Response:

Thank you so much. The reviewer is correct. We considered this valid concern. Authors carefully considered these important comments. We took a long time and extensively reviewed, edited and revised these bolded technical errors. Here we found 2 important comments.

1.NMR: You are perfect. What you are suggesting is the standards. NMR should be per 1000 live births. The reviewer is correct. What you are suggesting is the standards. We appreciate this valid observation. Normally, Neonatal mortality can expressed 1000 per live births Rate (NMR=per 1000 live births). However, some times, in small-scale research, we can estimate these neonatal death in “magnitude” or “proportion” to show the prevalence. Based on your recommendation, we corrected this valid comment, and corrected on page 3, line 61-64.

2. Incorrect citation: Thanks a lot. Dear reviewer, we are very sorry! This misused citation was unintentionally introduced during manuscript drafting. Now, we critically considered, and revised our endnote library to solve the issues. The newly modified changes were highlighted with red font color in the clean revised main manuscript.

4.-Furthermore, I suggest introducing shortly the Apgar-score as well as ANC and ANC follow up in the introduction. You cannot expect that the international readership knows this words/abbreviations.

Authors' response:

Thank you a lot. We appreciated your valid observation. Now authors considered your valid concern and explicitly incorporated your input in the operational definition (in the methods) (APGAR Score, ANC and ANC follow up) and the newly modified changes were highlighted with red font color in the clean revised manuscript.

METHODS

5.-In the study population sample size, data collection and data analysis are presented carefully. However, the sample size calculation can be presented more condensed. And the coding of neonatal mortality can be dropped from the explanation.

AUTHORS' RESPONSE:

Thank you a lot for such a technical input. Now, authors considered the raised issue. After thoroughly and critically revised this important comment, we removed previous bulky and long sentence and replaced with newly corrected one. 

6.Furthermore, under analysis it was mentioned that collinearity analysis was done, but the results were not presented. I expect a high collinearity between apgar-score and asphyxia (which is defined via apgar score) and ANC follow up and No of ANC-visits. Therefore, the results are important to present.

AUTHORS' RESPONSE:

Great, thanks a lot for such important intellectual input. These are very critical and concerns issues in the final models of multivariable analysis. Dear, reviewer, really we appreciate your insightful and logical input. You are perfect. Usually we expect similar predictor variables to have collinearity effect(highly correlation). For example,( primiparity Vs multiparity, Apga score Vs Birth asphyxia, Grand multipara Vs Advance maternal etc..). However, in our case, we didn’t encounter any collinearity effect between “Apgar score” and “Birth asphyxia” in using both VIF and Tolerance. All VIF results were less than 5 and All tolerance results were greater than 0.1. For instance, in our case= 

1.Apgar score (VIF= 1.44, Tolerance=0.70), 2.Ashyxia(VIF= 1.024, Tolerance=0.97) 

3.ANC follow(VIF=1.012, Tolerance=0.98)

7.Result presentation is nice and carefully done.

Authors response: 

Thank you so much. Authors greatly acknowledge for your countless effort.

8.Table 2: I suggest to present for binary variables only one line. If you know that 8.6 (n=44) have Hypothermia, then it is implicitly given that the rest do not have hyperthermia this line can be dropped from the table. However, that is relevant only for binary/dichotomous variables.

AUTHORS' RESPONSE:

Great, Thanks a lot for such implicit and critical review for our premature paper. Dear reviewer, you are perfect. We really appreciate this valid observation. However, in our case, the questionnaire/tool was prepared in binary response to assess neonatal conditions(Yes or No). Here, our intention is assess presence or absence of hypothermia. Thus. we prepared questions for hypothermia as: Does the neonate have Hypothermia at admission? 1. Yes 2. No, However, there was no question to assess hyperthermia. Thank a lot for your insightful suggestion and recommendation. 

9.On the other hand, there is one variable (number of ANC visits) which is not dichotomous/binary, but only presented in this form, clarification is necessary.

AUTHORS' RESPONSE:

Thanks a lot. Number of ANC visits were presented in Table 2 as 1. 1-3 visits, 2.>3visits….However, we didn’t consider it for regression analysis, instead we selected Presence or absence of ANC follow up. Actually, this is very important concern.

10.Table 4: significant results are only marked under AOR but not under COR. Please add this. Furthermore, both abbreviations need to be introduced under the table.

AUTHORS' RESPONSE:

Thanks a lot. You are perfect. These are very important observation. These valid comments and suggestions were addressed based on your insightful suggestion in both tables 3 and 4. 

11.Figure 1. The arrow text line “Simple random sampling (SRS)” can be dropped. The line above contains 196+178+140 which gives exactly 514. No random sampling took place at this stage of sampling procedure.

AUTHORS' RESPONSE:

Thank you very much. In fact, this is a valid concern. We critically considered this point, and dropped arrow text line from fig1 based on your valid suggestion. Thank you.

12.Figure 2: Can be deleted, The results of this figure can be added in table 2

AUTHORS' RESPONSE:

Great! Thanks a lot. Since it is pie-chat, the binary outcome is not suitable for this figure. Thank you for this intellectual input. Based on your insightful suggestion and your co-reviewers, we modified this figure from previous pie-chart to simple bar-chat. Since it is the major finding of our study, All authors critically discussed on this issue, and decided to change the figure 2 to simple bar-chart for simplicity of the readers(for easily capturing of the magnitude of NM).

13.The discussion is nicely done, good work.

AUTHORS' RESPONSE:

Thank you so much. Authors acknowledged you for your countless effort to review our paper.

NB: New notification for Reviewer 3

Dear, reviewer, thank you very much for your countless effort. Finally, we kindly notify you that during revision of our manuscript, we revised the final model of multivariable analysis and we checked one predictor variable (Birth weight of newborn). The BW was categorized as:

1.LBW= less than 2500mg , 2.Normal BW= 2500-4000mg

3.Macrosomia: >4000mg. In SPSS analysis, initially we used the “Macrosomia” as Ref. category, and no significant association was found in multivariable analysis. However, in current revision, the authors critically revise this issue and shifted/substituted the previous Ref. category (>4000mg) to “Normal BW(2500-400mg)”, considering the remaining two categories as exposure. As a result, LBW becomes significant predictor of Neonatal mortality. For further details, we submitted SPSS data used for analysis to the Journal as Supplementary File 1(S1). We highlighted newly modified changes with red font color in the Table 3 and 4, and elsewhere in the text. Thank you very much for your time and consideration.

End of authors’ responses to Reviewer 3

---

## [Decision Letter · Decision Letter 1]

18 Oct 2021

PONE-D-21-13372R1Neonatal mortality and Associated Factors among Neonates Admitted to Neonatal Intensive Care Unit in Public Hospitals of Somali Regional State, Eastern Ethiopia: A two years’ retrospective AnalysisPLOS ONE

Dear Dr. Kure,

Thank you for submitting your manuscript to PLOS ONE. After careful consideration, we feel that it has merit but does not fully meet PLOS ONE’s publication criteria as it currently stands. Therefore, we invite you to submit a revised version of the manuscript that addresses the points raised during the review process.

Thank you for your response to the reviewer comments. The reviewers consider the manuscript to be much improved. However, some additional editorial concerns must be addressed before the manuscript can be considered for publication. Specifically, we found a degree of text overlap between your submission and the following previously published works:In lines 55-60, there is some overlap with http://www.kma.org.kw/uploads/versions/EEQGAXUUOAQIGLDPHVORFANO.pdf (pages113-114)In lines 81-85 there is some overlap with https://documents1.worldbank.org/curated/en/384721537219780286/pdf/129971-AR-PUBLIC-UN-IGME-Child-Mortality-Report-2018.pdf?fbclid=IwAR2X1_3wyyZFl_v1-6yrBvZgIM0jMjnw4cJL4JxRBou8XV_eJfo2oGOyg20 (page 12)

Please review the entire manuscript, especially the abovementioned sections to ensure that you rephrase any duplicated text and cite your sources in full. In addition, please review the manuscript again to check for any remaining typographical and grammatical errors, particularly in the Abstract.

Thank you for your attention to these requests.

We look forward to receiving your revised manuscript.

Kind regards,

Marianne Clemence

Associate Editor

PLOS ONE

Journal Requirements:

Additional Editor Comments (if provided):

Reviewers' comments:

Reviewer's Responses to Questions

**Comments to the Author**

1. If the authors have adequately addressed your comments raised in a previous round of review and you feel that this manuscript is now acceptable for publication, you may indicate that here to bypass the “Comments to the Author” section, enter your conflict of interest statement in the “Confidential to Editor” section, and submit your "Accept" recommendation.

Reviewer #1: All comments have been addressed

Reviewer #2: All comments have been addressed

2. Is the manuscript technically sound, and do the data support the conclusions?

Reviewer #1: Yes

Reviewer #2: Yes

3. Has the statistical analysis been performed appropriately and rigorously? 

Reviewer #1: Yes

Reviewer #2: Yes

4. Have the authors made all data underlying the findings in their manuscript fully available?

Reviewer #1: Yes

Reviewer #2: Yes

5. Is the manuscript presented in an intelligible fashion and written in standard English?

Reviewer #1: Yes

Reviewer #2: Yes

6. Review Comments to the Author

Reviewer #1: Significantly improved and have addressed most of my concerns.

I would suggest one further review with a language editor as there are still a few syntax errors, and then after that to accept for publication.

Reviewer #2: (No Response)

7. PLOS authors have the option to publish the peer review history of their article (what does this mean?). If published, this will include your full peer review and any attached files.

Reviewer #1: No

Reviewer #2: **Yes: **Dr Hedia Bellali, Associate professor in Epidemiology and Public Health, Medical Faculty of Tunis, Tunisia

---

## [Author Response · Author response to Decision Letter 1]

23 Oct 2021

Authors’ response to the editor’s minor comments and suggestions:

• Manuscript ID: PONE-D-21-13372

• MS title: Neonatal mortality and Associated Factors among Neonates Admitted to Neonatal Intensive Care Unit in Public Hospitals of Somali Regional State, Eastern Ethiopia: A multicenter retrospective Analysis

• Authors: Hamda Ahmed Mohamed1, Zemenu Shiferaw2, Abdurahman Kedir Roble3, Mohammed Abdurke Kure4

• Journal’s name: PLOS ONE

• Date: October 20, 2021

Dear editor(s), your Excellency!

First and foremost, we would like to thank PLOS ONE Journal’s editorial office for giving us an opportunity to submit a revised draft of our manuscript entitled “Neonatal mortality and Associated Factors among Neonates Admitted to Neonatal Intensive Care Unit in Public Hospitals of Somali Regional State, Eastern Ethiopia: A multicenter retrospective Analysis” to this high visibility impact factor and peer reviewed Journal. We appreciate the time and effort that you and the reviewers dedicated to providing feedback on our manuscript. We are also very grateful for the insightful comments and suggestions to our paper. We have incorporated minor comments made by handling editor. Newly modified changes were highlighted in red font color in the specific part of the revised manuscript. 

Subject: Authors’ Response to Specific Comments and Suggestion of Academic Editor

To: Handling Editor(s)

From: Mohammed Abdurke Kure (Corresponding Author)

Above all, authors thank you for your insightful constructive comments and suggestions that helped us to improve and enrich our manuscript to this status. Next, here in the table below, we tried to address your valuable comments, suggestions and concerns one by one. 

Editor’s Comments and suggestion to the Authors 

Editor’s General Comments and Suggestions 

Authors’ Responses: Overall, thank you so much for your cooperation to handle our manuscript. Handling paper is really needs dedication and strong commitment. Thanks a lot!

1.Thank you for submitting your manuscript to PLOS ONE. After careful consideration, we feel that it has merit but does not fully meet PLOS ONE’s publication criteria as it currently stands. Therefore, we invite you to submit a revised version of the manuscript that addresses the points raised during the review process. 

Authors’ Responses: Great! We are very grateful for the entire editorial office in general, and respective editorial team in particular. 

• Further, above all, we would like to extend our deepest appreciation to the handling academic editor, for the time and countless effort that she/he dedicated to handle our paper as academic editor (handling editor). We understand that handling paper is really needs dedication and strong commitment. 

• Finally, authors thank you for giving us an opportunity to submit our revised manuscript to such legitimate and high visibility impact factor Journal (PLOS ONE). 

2. Thank you for your response to the reviewer comments. The reviewers consider the manuscript to be much improved. However, some additional editorial concerns must be addressed before the manuscript can be considered for publication. 

Authors’ Responses: Ok, we accepted with thanks. We appreciate the time and effort that you and the reviewers have dedicated to providing feedback on our manuscript. 

• We critically revised all editor’s concerns and comments, and we addressed all accordingly(on respective page numbers). 

3. Specifically, we found a degree of text overlap between your submission and the following previously published works:

 a. In lines 55-60, there is some overlap with http://www.kma.org.kw/uploads/versions/EEQGAXUUOAQIGLDPHVORFANO.pdf (pages113-114)

 b. In lines 81-85 there is some overlap with https://documents1.worldbank.org/curated/en/384721537219780286/pdf/129971-AR-PUBLIC-UN-IGME-Child-Mortality-Report-2018.pdf?fbclid=IwAR2X1_3wyyZFl_v1-6yrBvZgIM0jMjnw4cJL4JxRBou8XV_eJfo2oGOyg20 (page 12) 

Authors’ Responses: Thanks a lot. Really, this is very critical concerns. Dear editor, we are very sorry for such unnecessary bolded mistake made in the previous last submission. In this regard, we observed two important suggestion:

 a. overlap issues of lines 55-60:

• Dear editor, these overlapping/similarity texts were unintentionally introduced during the 2nd round of revised submission. It’s all our fault and therefore, we sincerely apologize for all these an unintentionally introduced technical errors. 

• These valid comments were extensively revised, and addressed based on your implicit and valuable suggestion. Newly changed and corrected parts were highlighted with red font color in the clean revised manuscript (please see page=3, lines: 54-64). Thank you once again. 

b. overlap issues of lines 81-85:

• Thank you so much: The editor is perfect. 

• Similarly, this valid observation was also critically revised and rephrased based on your insightful suggestions. The newly modified change were highlighted with a red-font color in the clean revised manuscript.(Please see, page 4 & 5, lines: 83-97). 

4. Please review the entire manuscript, especially the abovementioned sections to ensure that you rephrase any duplicated text and cite your sources in full. In addition, please review the manuscript again to check for any remaining typographical and grammatical errors, particularly in the Abstract.

Authors’ Responses: Thank you very much. In fact, this is very important suggestion. We are very grateful for this implicit comment and suggestion. Thank you so much for such a valuable intellectual input. Authors critically considered this valid input. 

• Accordingly, we thoroughly revised and edited not only the abstract, but also the whole parts of our manuscript.

• In addition, we extensively corrected all copy-editing errors made in the previous submission. Authors also sent the manuscript to language expert/editor who critically reviewed, edited and corrected all language related errors made in a submitted manuscript. 

• Finally, the newly modified change were highlighted with red font color in the clean revised manuscript (Page 2). 

Authors’ Responses: Thank you so much. You are perfect. The issues of reference are very critical, and have to be revised and corrected before any decision for the acceptance of the Manuscript for publication. 

• Even it is the authors mandatory to critically review the references styles and its relevancy (Both in text-citation and Bibliography) to avoid any inconsistence and wrong citations across the document. 

• We took a long time, thoroughly and critically revised our endnote library. Moreover, any wrong citations were checked in the word document and refined accordingly. 

• Further, in the document citation, we checked the references for any repetition, incompleteness, and inconsistences. 

• Finally, we addressed these important concerns based on your insightful suggestions. Newly corrected references were highlighted with red font color in the clean revised manuscript.

In summary:

Overall, thank you all for your unquantified effort to enrich this manuscript by forwarding your insightful intellectual input. We are very grateful for all reviewers and the entire editorial office in general and respective academic editor in particular. Dear editor, we learned a lot from all steps of review process made in this manuscript. Really, publications process is learning forum. 

=All we can is thanks!!

End of authors’ responses for Handling Editor(s)

---

## [Decision Letter · Decision Letter 2]

4 May 2022

Neonatal mortality and Associated Factors among Neonates Admitted to Neonatal Intensive Care Unit in Public Hospitals of Somali Regional State, Eastern Ethiopia: A multicenter retrospective Analysis

PONE-D-21-13372R2

Dear Dr. Kure,

We’re pleased to inform you that your manuscript has been judged scientifically suitable for publication and will be formally accepted for publication once it meets all outstanding technical requirements.

Kind regards,

George Vousden

Deputy Editor-in-Chief

PLOS ONE

Additional Editor Comments (optional):

Reviewers' comments:

Reviewer's Responses to Questions

**Comments to the Author**

1. If the authors have adequately addressed your comments raised in a previous round of review and you feel that this manuscript is now acceptable for publication, you may indicate that here to bypass the “Comments to the Author” section, enter your conflict of interest statement in the “Confidential to Editor” section, and submit your "Accept" recommendation.

Reviewer #2: All comments have been addressed

2. Is the manuscript technically sound, and do the data support the conclusions?

Reviewer #2: Yes

3. Has the statistical analysis been performed appropriately and rigorously? 

Reviewer #2: Yes

4. Have the authors made all data underlying the findings in their manuscript fully available?

Reviewer #2: Yes

5. Is the manuscript presented in an intelligible fashion and written in standard English?

Reviewer #2: Yes

6. Review Comments to the Author

Reviewer #2: The authors have adressed all the reviewer's comments. However, the paper layout is still to be improved, there are some mistakes of presentation: spaces between references and text...

7. PLOS authors have the option to publish the peer review history of their article (what does this mean?). If published, this will include your full peer review and any attached files.

Reviewer #2: **Yes: **Prof Hedia Bellali

---

## [Editor Report · Acceptance letter]

19 May 2022

PONE-D-21-13372R2 

*Neonatal mortality and Associated Factors among Neonates Admitted to Neonatal Intensive Care Unit at Public Hospitals of Somali Regional State, Eastern Ethiopia: A multicenter retrospective Analysis*  

Dear Dr. Kure:

I'm pleased to inform you that your manuscript has been deemed suitable for publication in PLOS ONE. Congratulations! Your manuscript is now with our production department. 

Kind regards, 

on behalf of

Dr. George Vousden 

Staff Editor

PLOS ONE